# DEMYSTIFYING REINFORCEMENT LEARNING IN AGENTIC REASONING

## ABSTRACT

Recently, the emergence of agentic RL has showcased that RL could also effectively improve the agentic reasoning ability of LLMs, yet the key design principles and optimal practices remain unclear. In this empirical study, we conduct a comprehensive and systematic investigation to demystify reinforcement learning in agentic reasoning from three key perspectives: *data, algorithm, and reasoning mode*. We highlight our key insights: (i) Replacing stitched synthetic trajectories with real end-to-end tool-use trajectories yields a far stronger SFT initialization; high-diversity, model-aware datasets sustain exploration and markedly improve RL performance. (ii) Exploration-friendly techniques are crucial for agentic RL, such as clip higher, overlong reward shaping, and maintaining adequate policy entropy could improve the training efficiency. (iii) A deliberative strategy with fewer tool calls outperforms frequent tool calls or verbose self-reasoning, improving tool efficiency and final accuracy. Together, these simple practices consistently enhance agentic reasoning and training efficiency, achieving strong results on challenging benchmarks with smaller models, and establishing a practical baseline for future agentic RL research. Beyond these empirical insights, we further contribute a high-quality, real end-to-end agentic SFT dataset along with a high-quality RL dataset, and demonstrate the effectiveness of our insights in boosting the agentic reasoning ability of LLMs across four challenging benchmarks, including AIME2024/AIME2025, GPQA-Diamond, and LiveCodeBench-v6. With our recipes, 4B-sized models could also achieve superior agentic reasoning performance compared to 32B-sized models.

## 1 INTRODUCTION

Beyond pre-training and supervised fine-tuning (SFT) stages, recent advancements in reinforcement learning (RL) (Schulman et al., 2017; Rafailov et al., 2023; Yang et al., 2025c;b; Shao et al., 2024; Wang et al., 2025c) have introduced a new scaling axis that aligns large language models' (LLMs) behavior to incentivize reasoning fidelity by encouraging the generation of effective chain-of-thought (CoT) trajectories. Building on this, the paradigm of agentic reasoning (Li et al., 2025a; Wu et al., 2025; Li et al., 2025c; Sun et al., 2025; Jin et al., 2025; Feng et al., 2025; Dong et al., 2025b) further empowers LLMs to move beyond self-contained generation, equipping them with the ability to integrate external tools throughout the reasoning process. This shift has unlocked remarkable progress across domains such as mathematics, scientific discovery, and code generation.

Despite the rapid growth of these advances, scaling RL for agentic reasoning remains challenging. Directly applying policy optimization methods such as GRPO (Shao et al., 2024; Guo et al., 2025) often leads LLM agents to suffer from suboptimal training and inference behaviors, including inefficient on-policy rollout sampling, reward & entropy collapse (Cui et al., 2025), and unstable training dynamics. This highlights the unsolved limitations from three perceptiveness:

**1** *Data wise.* Current data curation pipelines often rely on stitch-style data synthesis (Feng et al., 2025; Schick et al., 2023), where segments of internal reasoning are manually replaced with tool outputs. Such patchwork overlooks the natural connectivity of reasoning and tool use, preventing the data from faithfully mimicking real multi-turn trajectories that indicate when and why tools should be invoked.

**2** *Algorithm wise.* Despite rapid progress in GRPO-based variants, the optimal RL recipe for agentic reasoning remains unclear. Existing methods differ in their optimization granularity (token-,

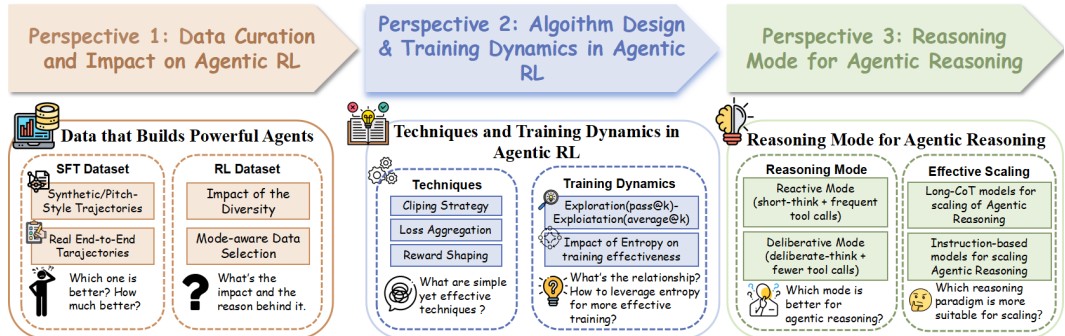

Figure 1: The overview of our empirical study.

sequence-, or trajectory-level) and impose distinct inductive biases: some encourage exploration by relaxing clipping Yu et al. (2025) or managing entropy (Wang et al., 2025b), while others suppress it through strong KL regularization (Cheng et al., 2025b) or conservative clipping. A principled understanding of when and how to deploy these algorithms is still missing.

③ *Reasoning Mode wise.* Open puzzles are unsolved regarding the allocation of turn budgets, the trade-off between response length and tool-call efficiency, and the impact of long-CoT predispositions on multi-turn reasoning. These uncertainties obscure the principles of agent reasoning modes, often leading to either overthinking (long, inefficient loops) or underthinking (premature tool reliance) during the agents' reasoning process.

The above challenges motivate us to perform a systematic investigation of recent studies along these three perspectives, aiming to identify key factors that either hinder or enhance agentic reasoning in LLM agents as shown in fig. 1. Specifically, we organize our paper as follows:

- In **Section 2 (to address ①)**, we analyze how data curation design and diversity affect SFT and RL training stages, respectively. We observe that training directly on synthetic trajectories fails to provide reliable signals for when and how to invoke tools, preventing agents from learning optimal integration points, and also lacks sufficient data diversity to encourage effective exploration. To address this, we curate real end-to-end SFT dataset and high-diversity, model-aware RL dataset that improve the training efficiency and agentic reasoning performance.

- In **Section 3 (to address ②)**, we compare GRPO-based RL algorithms and find that conservative clipping and KL divergence penalty overly constrain exploration during training. In addition, we analyze the roles of pass@k and average@k as guiding metrics, revealing how they capture the exploration–exploitation trade-off and highlight performance bottlenecks. We further show that sustaining higher entropy, especially for weaker models, is the key to improving RL efficiency.

- In **Section 4 (to address ③)**, we investigate reasoning-mode components such as the number of tool calls and overall response length, and their relationship to performance. We find that fewer, more deliberate tool interactions often yield better results, showing that over-reliance on external calls does not necessarily improve performance and that the key lies in effective and accurate tool invocations integrated into the model's agentic reasoning process.

## 2 DATA IN AGENTIC REASONING

This section empirically examines how data affects agent training, comparing real end-to-end agentic vs synthetic stitch-style trajectories for cold-start SFT and evaluating high-diversity and model-aware datasets designed to maintain exploration to achieve effective RL.

### 2.1 REAL END-TO-END TRAJECTORIES VERSUS SYNTHETIC STITCH-STYLE TRAJECTORIES

**Motivation** Current agentic training pipelines often rely on LLM-edited or template-based synthetic trajectories, which replace selected reasoning steps with tool invocations like ReTool Feng et al. (2025). While scalable, such stitch-style data inevitably misses critical decision cues: not only how to call a tool, but also when, why, and what to do next. This raises the question: do real end-to-end trajectories provide qualitatively richer learning signals and stronger initialization for RL?

**Setup.** For the synthetic baseline, we directly adopt the multi-turn SFT dataset from ReTool (Feng et al., 2025), where challenging long-CoT steps are substituted with tool invocations and responses.

Table 1: Comparison between the impact of our curated real end-to-end SFT dataset and the synthetic SFT dataset on AIME 2024 and AIME 2025.

| Dataset & Metric | Qwen2.5-7B-Instruct | | Qwen3-4B-Instruct-2507 | |
|---|---|---|---|---|
| | w/ synthetic trajectory | w/ **real** trajectory | w/ synthetic trajectory | w/ **real** trajectory |
| **AIME 2024** | | | | |
| average@32 | 6.77% | **17.91%** (+11.14%) | 4.38% | **33.23%** (+28.85%) |
| pass@32 | 42.11% | **57.57%** (+15.46%) | 35.15% | **75.66%** (+40.51%) |
| maj@32 | 10.50% | **27.13%** (+16.63%) | 0.01% | **51.64%** (+51.63%) |
| **AIME 2025** | | | | |
| average@32 | 5.21% | **18.24%** (+13.03%) | 3.65% | **29.79%** (+26.14%) |
| pass@32 | 25.56% | **48.42%** (+22.86%) | 22.22% | **72.88%** (+50.66%) |
| maj@32 | 12.08% | **29.18%** (+17.10%) | 0.10% | **45.82%** (+45.72%) |

For real end-to-end trajectories, we use our curated dataset mentioned in appendix B.1. We fine-tune Qwen2.5-7B-Instruct and Qwen3-4B-Instruct-2507 on both datasets (real vs. synthetic) under identical settings, and evaluate agentic reasoning on AIME2024/AIME2025.

**Result.** We evaluate using average@32 (overall agent performance), pass@32 (ability boundary (Deng et al., 2025)), and maj@32 (performance stability). As shown in table 1, real trajectories deliver a clear improvement: Qwen3-4B-Instruct-2507 trained on real data achieves 29.97% on average@32, 72.88% on pass@32, and 45.22% on maj@32 on AIME2025. In contrast, the synthetic baseline yields below 10% on average@32 with unstable performance and a significantly lower ability upper bound. Thus, real trajectories establish a much stronger and more stable starting point for RL. Unless otherwise stated, we use SFT checkpoints trained on our curated real agentic dataset, denoted as **Qwen3-4B-RA-SFT** and **Qwen2.5-7B-RA-SFT**.

**Analysis.** The superiority of real trajectories lies in their ability to capture **complete agentic reasoning behaviors** through real end-to-end reasoning processes that synthetic stitching cannot replicate. Specifically, our dataset preserves: (i) **pre-call analysis**, localizing which subproblems are efficiently solved via tools; (ii) **guarded execution**, with intermediate checks; (iii) **error recovery and strategy revision**, after failed attempts; (iv) **self-reflection and calibration**, before invoking tools.

> **Takeaway 2.1: Curating Agentic SFT Data**
>
> Real agentic trajectories with coherent and end-to-end tool-use behaviors can not only teach the agent to use tools but it also scale the ability boundary and produce more stable reasoning, while synthetic trajectories fail.

## 2.2 DIVERSE DATA MAINTAINS HIGH ENTROPY IN TRAINING

**Motivation.** Most existing works (Feng et al., 2025; Shang et al., 2025; Li et al., 2025d; Dong et al., 2025a) focus on purely mathematical datasets for RL training, aiming to enhance problem-solving ability on reasoning-heavy benchmarks. While intuitive, this narrow scope overlooks a critical factor: **dataset diversity**. Prior discussions of diversity in multi-task RL (Havrilla et al., 2024; Shen et al., 2025) only emphasize outcome-level benefits. However, how diversity influences the training dynamics, especially policy entropy and exploration efficiency, remains underexplored.

**Setup.** We construct our RL dataset with higher diversity in appendix B.2, for comparison, we choose DAPO-Math-17k as the baseline. We utilize GRPO-TCR (section 3) to fine-tune Qwen3-4B-RA-SFT and Qwen2.5-7B-RA-SFT under identical hyperparameters and training budgets.

**Result.** As shown on the left in fig. 2, training with the diverse dataset leads to significantly higher entropy gain during the early stage and sustains this entropy at a higher level throughout convergence. This indicates that diverse data directly drives richer exploration behaviors. Moreover, in the right figures of fig. 2, we observe faster and more efficient learning: with our diverse dataset, the agent achieves over 50% average@32 accuracy on AIME2025 within only 150 steps, while the DAPO-Math baseline requires 220 steps to reach the same level.

**Analysis.** We interpret entropy as a proxy for exploration breadth: higher entropy means the policy continues to consider diverse reasoning paths rather than prematurely collapsing to a narrow deterministic strategy (We will discuss the entropy mechanism in detail in section 3.2 and section 3.3).

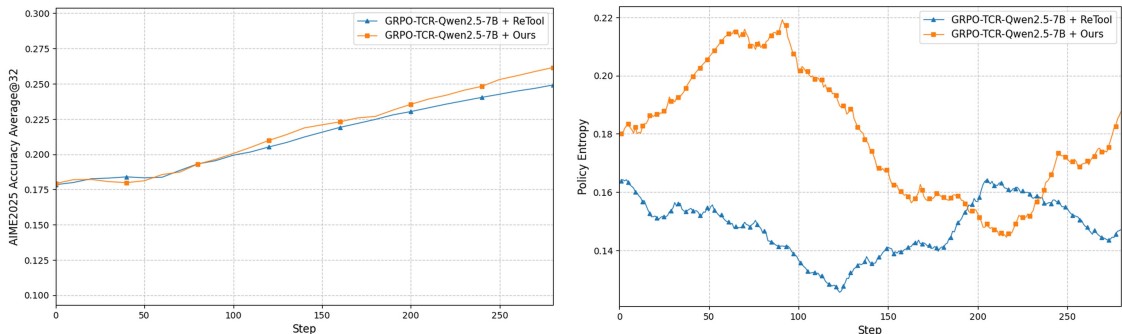

Figure 2: Comparison between our dataset with higher diversity and the ReTool dataset, which only contains math problems. **Left** is the average@32 accuracy on AIME2025 during training based on two different dataset. **Right** is the policy entropy during the training process.

Thus, dataset diversity not only improves outcome metrics but also **reshapes the training dynamics** by maintaining exploration capacity, making RL both faster and more stable.

---

**Takeaway 2.2: Dataset Construction for RL training**

Diverse RL datasets sustain higher policy entropy, directly incentivizing broader exploration and yielding faster, more stable agentic RL training.

---

### 2.3 MODEL-AWARE DATASETS FOR MORE EFFECTIVE RL

**Motivation.** During training, we observed a clear divergence between two models of different capacity: Qwen3-4B-Instruct-2507 exhibited consistent and sustained policy improvement, while Qwen2.5-7B-Instruct failed to improve despite identical algorithms and datasets, rapidly encountering a bottleneck. Specifically, the average reward of Qwen2.5 stagnated around zero, while Qwen3 consistently achieved positive rewards. This illustrates a **competence–difficulty mismatch**: when the base policy is too weak relative to the dataset, it cannot extract meaningful gradients for policy updating. To address this issue, we construct a **model-aware RL dataset** that adapts the task distribution to the capacity of the model.

**Setup.** We use our SFT model to perform 8 rollouts per problem on the 30k RL dataset, taking the proportion of correct solutions as a proxy for problem difficulty with respect to the given model. After trajectory verification, we discard trivial problems with 0% or 100% accuracy (which provide no learning signal) and label the remainder with three difficulty levels: easy (accuracy $\geq 0.75$), medium ($0.75 >$ accuracy $> 0.25$), and hard (accuracy $\leq 0.25$). Since the Qwen3 model already shows effective training on the full dataset, we use its empirical difficulty histogram as the target distribution. Based on this distribution, we curate a model-aware dataset tailored for Qwen2.5 and retrain it using the same GRPO-TCR algorithm and hyperparameters as in the original setting.

**Result.** As shown in fig. 3, training with the curated model-aware dataset yields significantly more effective improvement than with the unfiltered dataset. Moreover, fig. 3 shows that the average reward rises substantially, producing stronger and more consistent gradient signals. Consequently, it provides more valid rewards for the computation of advantage, amplifying the gradient signals leading to more effective and stable RL training. After breaking the performance bottleneck, we can collect the model-aware dataset based on its current ability for more effective training.

---

**Takeaway 2.3: Data Selection for RL training**

Model-aware data provides stronger gradient signals, amplifying learning feedback to overcome weak-model performance bottlenecks and improve RL training efficiency.

---

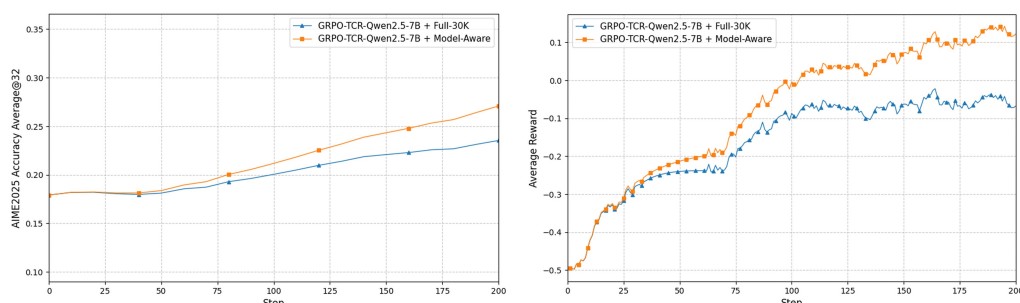

Figure 3: The comparison and analysis between the impact of the 30k full dataset and our tailored dataset for Qwen2.5-RA-SFT on subsequent RL training. **Left** is the average@32 performance on AIME2025. **Right** is the analysis for the average reward during training.

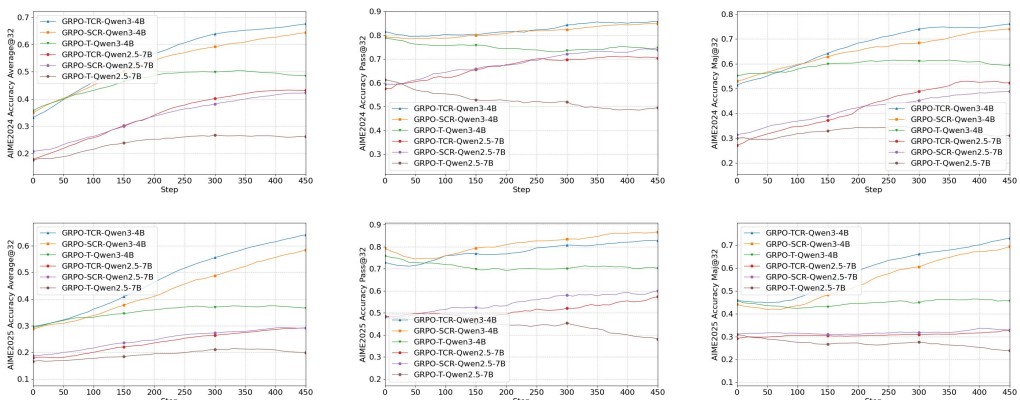

Figure 4: The overall performance of our constructed three recipes: **GRPO-T**, **GRPO-TCR**, and **GRPO-SCR** on AIME2024/AIME2025 benchmark.

## 3 ALGORITHMIC DESIGN AND TRAINING DYNAMICS IN AGENTIC RL

Recent advancements in RLVR for reasoning LLMs, especially GRPO-based methods (Yu et al., 2025; Dong et al., 2025b; Zheng et al., 2025; Zhao et al., 2025; Liu et al., 2025a;b) have demonstrated that there are many possible techniques that could further enhance policy optimization. Other works (Cui et al., 2025; Deng et al., 2025; Agarwal et al., 2025; Chen et al., 2025) focus on exploring the training dynamics (e.g., entropy and pass@k) to explain why and how these improvements could emerge. For agentic RL, however, it remains unclear (i) what techniques work best for policy optimization, (ii) what is the relationship between the exploration(pass@k)-exploitation(average@k), and (iii) how does entropy affect training effectiveness, stability, and final performance.

### 3.1 THE IMPACT OF THE RLVR TECHNIQUES ON AGENTIC RL

**Recipe Design** In this section, we investigate three key techniques for agentic RL: **Loss Aggregation Granularity**, **Reward Shaping**, and **Clipping Strategy** as mentioned in appendix A.2. We first construct two improved GRPO recipes that combine multiple RLVR techniques. For the first recipe **GRPO-TCR**, we incorporate **T**oken-level loss, **C**lip higher and overlong **R**eward shaping techniques with GRPO. For the second recipe **GRPO-SCR**, we incorporate **S**equence-level loss, **C**lip higher, overlong **R**eward shaping techniques with GRPO. For the baseline, we adhere to the implementation in (Shao et al., 2024), and change the sample-level loss to token-level, we hereby denote this recipe as **GRPO-T**. Based on the experiment results, we aim to pinpoint **simple yet effective** techniques that deliver consistent improvements in final performance and training efficiency.

**Setup.** Here we utilize the hyperparameters and reward functions mentioned in appendix A.2 to train both Qwen3-4B-RA-SFT and Qwen2.5-7B-RA-SFT based on our complete RL dataset to compare the training dynamics of different recipes. For GRPO-TCR, the $\epsilon_{\text{high}}$ is set to 0.28, and $\epsilon_{\text{low}}$ is set to 0.20, for GRPO-SCR, the $\epsilon_{\text{high}}$ is set to 0.0004 and $\epsilon_{\text{low}}$ is set to 0.0003. For GRPO-T, the $\epsilon$ is set to 0.20 and the reward function is $r_{\text{out+tool}}$. For both recipes that incorporated with overlong reward shaping, the reward is denoted as $r_{\phi} = r_{\text{out+tool}} + r_{\text{length}}$. Specifically, we also use three

metrics to comprehensively evaluate the impact of the applied RLVR techniques on AIME2024 and AIME2025 benchmarks.

**Result.** First, we compare **GRPO-TCR** and **GRPO-T** to investigate the impact of **clip higher** and **overlong reward shaping** on Agentic RL. Specifically, for Qwen3-4B-RA-SFT, GRPO-TCR has achieved remarkable improvements compared to GRPO-T on AIME2024/AIME2025. It achieves **70.93%/68.13%** with an initial accuracy of 29.79% and 33.23% on average@32 metric within only 450 steps. In contrast, GRPO-T only achieves the best average@32 performance of 54.7%/ 40.93% on AIME2024/AIME2025, which GRPO-TCR could achieve within only 100 training steps, utilizing only **25%** of the training computation of GRPO-T. The results indicate that simply **applying clip higher and overlong reward shaping techniques could effectively improve the agentic reasoning performance and the efficiency of agentic RL.**

Then, we compare **GRPO-TCR** and **GRPO-SCR** to investigate the impact of **loss aggregation granularity** on agentic RL. We observe that for Qwen2.5-7B-RA-SFT, which has weak initial performance and exploration capabilities, token-level loss and sequence-level loss achieve comparable average@32 performance on AIME2024/2025. However, for Qwen3-4B-RA-SFT, which has stronger initial performance and exploration capabilities, token-level loss consistently outperforms sequence-level loss in terms of convergence speed and peak accuracy. Specifically, token-level loss exceeds sequence-level loss by 3.95% on AIME24 and 3.86% on AIME25 under the same training budget. It is because the token-level loss ensures each token contributes equally to the optimization signal, thereby leveraging the model's exploratory capacity more effectively. This suggests that **token-level loss could improve training efficiency and agentic reasoning ability compared to sequence-level loss for models with better initial performance and exploration ability**.

---

**Takeaway 3.1: The Effective Techniques for Agentic RL**

1. Clip higher and overlong reward shaping are simple yet effective techniques to improve the performance of Agentic RL.
2. Token-level loss outperforms sequence-level loss when the models have better exploration ability in convergence speed, peak accuracy, and training robustness.

---

### 3.2 EXPLORATION–EXPLOITATION DYNAMICS IN AGENTIC RL

**Motivation.** Prior studies (Chen et al., 2025; Deng et al., 2025) show that in conventional RL, most gains in the *pass@k* metric come from the SFT stage, where diverse external solutions expand the model's ability bound. Subsequent RL training primarily strengthens existing internal solutions, yielding a more deterministic policy that improves exploitation (Pass@1) but often suppresses further exploration (Pass@k). However, this characterization is largely based on a **self-contained generation process**, where the model relies solely on its internal capacity. In contrast, **agentic RL** fundamentally changes this dynamic: the model actively interacts with external tools during reasoning, learning not just to refine internal solutions but to **optimize its ability to explore, select, and exploit external resources**. This opens the question of whether the classical exploration–exploitation trade-off still holds, or whether agentic RL enables a qualitatively different trajectory where exploration is maintained or even amplified through tool use.

**Observation.** Our experiments in fig. 4 show that in agentic RL, both GRPO-TCR and GRPO-SCR achieve substantial and simultaneous improvements in pass@k and average@k (over 10% gains on AIME2024/AIME2025). However, this improvement does not hold unconditionally: with the baseline GRPO-T, we still observe the conventional trade-off where exploration is suppressed during training. We attribute this to the overly conservative design of GRPO-T: the combination of a restrictive clip upper bound and strong KL-regularization creates severe constraints on distribution shift, forcing the model to maintain self-contained generation patterns and preventing it from fully leveraging tool interactions.

**Analysis.** As we mentioned above, the multi-turn interactions with tools in agentic reasoning also introduce external information during training. The information from the external tools enables models to "think smarter" than purely "think longer" by developing more advanced cognitive abilities that autonomously utilize the tools to reason more efficiently, and learn from the feedback signals. Consequently, incentivizing these abilities through agentic reinforcement learning recipe

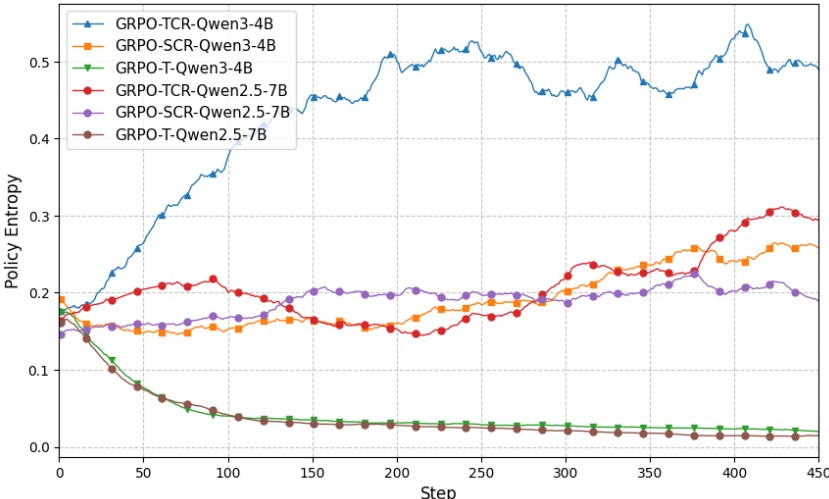

Figure 5: The analysis for the policy entropy in agentic RL training.

like GRPO-TCR and GRPO-SCR helps to further improve the pass@k performance and lead to higher ability bound for more effective training and better average@k performance.

What's more, we also observe that **the gap between average@k and pass@k emerges as a critical bottleneck for training efficiency.** RL training can be interpreted as a process of progressively **converting the model's pass@k performance into actual average@k gains**, with the achievable improvement bounded by an intrinsic ceiling determined by this gap. This perspective highlights that not only the absolute level of pass@k but also the magnitude of the average–pass discrepancy governs how much exploration can be effectively transformed into exploitation during training.

---

**Takeaway 3.2: Pass@k and Average@k in Agentic RL training.**

1. With external tool interactions, agentic RL can jointly improve pass@k and average@k while conventional RL failed.
2. The critical bottleneck for training efficiency is the gap between pass@k and average@k

---

### 3.3 WHEN HIGH ENTROPY DRIVES BETTER EFFICIENCY

**Motivation.** Recently, entropy has become a central signal in RL research, yet prescriptions diverge: some advocate minimizing it for more deterministic policies (Agarwal et al., 2025; Cheng et al., 2025b), while others exploit high-entropy tokens to foster exploration and avoid early collapse (Cui et al., 2025; Wang et al., 2025b;a). These views largely arise from conventional RL. In agentic RL, ARPO (Dong et al., 2025b) observes entropy spikes after tool calls and leverages them via adaptive rollouts, implying that tool-call steps induce useful uncertainty. This raises a central question: is higher/lower entropy generally beneficial, or is there an optimal range beyond which training destabilizes?

**Observation.** We investigate by visualizing entropy trajectories and relating them to training efficiency and reasoning performance. As shown in fig. 4 and fig. 5, GRPO-T exhibits an early entropy collapse, whereas the entropy for models (trained with GRPO-TCR and GRPO-SCR) with better performance rises faster and stabilizes at a higher level. This suggests that **greater policy entropy is associated with more effective agentic RL training and stronger agentic reasoning, which is not aligned with the entropy minimization theory in conventional RL**. Motivated by our observation, and noting that $\epsilon_{high}$ controls the exploration budget, we utilize different $\epsilon_{high}$ to test for an optimal entropy regime under identical training conditions.

**Setup.** To investigate the optimal entropy regime, we conduct experiments with different clip upper bounds, including 0.28,0.315,0.35 for GRPO-TCR and keep all other settings the same and train Qwen2.5-7B-RA-SFT and Qwen3-4B-RA-SFT. We report the three metrics including average@32, pass@32, and maj@32 to comprehensively evaluate when high entropy could improve training efficiency and when high entropy would lead to the collapse of the agentic reasoning performance.

**Results.** As shown in fig. 7, we observe a non-monotonic relation between the clip upper bound $\epsilon_{\text{high}}$ and training efficiency. Specifically, for Qwen2.5-7B-RA-SFT, modestly increasing $\epsilon_{\text{high}}$ (e.g., $0.28 \to 0.315$) accelerates performance improvements. It also improves learning efficiency across both models. For example, with $\epsilon_{\text{high}} = 0.315$, we achieve equivalent performance 40% faster, reaching the same results at step 60 that would otherwise require 100 steps when $\epsilon_{\text{high}} = 0.28$. However, pushing it further yields diminishing returns. For example, when we train Qwen3-4B-RA-SFT with a higher $\epsilon_{\text{high}} = 0.35$ it leads to worse training effectiveness despite a faster initial lift compared to a lower $\epsilon_{\text{high}}$, which is set to 0.28. In summary, a higher $\epsilon_{\text{high}}$ expands the exploration budget and improves short-horizon progress, yet overly aggressive clipping eventually slows convergence, introduces excessive entropy, which will lead to suboptimal agentic reasoning performance. It also indicates that when the entropy becomes too high, it will also lead to instability in training.

> **Takeaway 3.3: Entropy as a Driver of Training Efficiency.**
>
> 1. Agentic RL requires balanced policy entropy, which avoids both excessive entropy (insta-bility) and insufficient entropy (premature convergence) for optimal training effectiveness.
> 2. Weaker models require larger clip upper bounds to escape the performance bottleneck, while stronger models demand tighter bounds to prevent over-exploration.

## 4 REASONING MODES IN AGENTIC RL

A central question in **agentic RL** is how an agent should allocate its reasoning budget between **internal inference tokens** and **external tool calls**. Should an effective agent rely on frequent tool interactions with minimal internal thinking, or invest more inference tokens in deliberate reasoning before acting? To address this, we characterize two regimes: (i) **tool-call scaling**, where the agent engages in many short-think rounds with frequent tool usage, and (ii) **internal reasoning scaling**, where the agent performs deeper reasoning before issuing fewer but more targeted tool calls. This section empirically investigates these reasoning modes and identifies which strategy leads to more efficient and effective agentic reasoning.

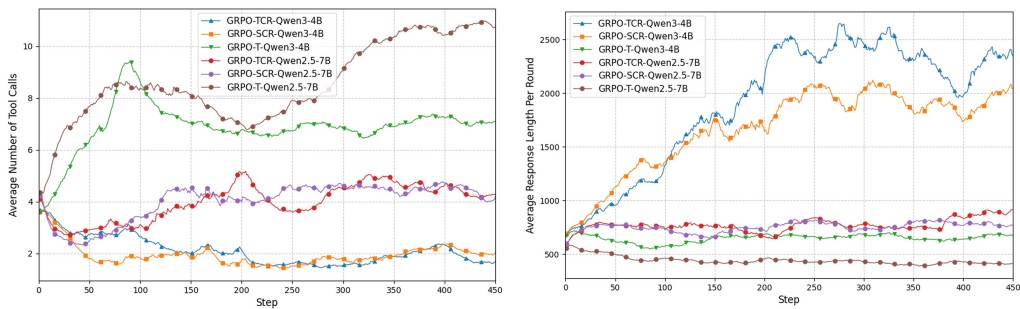

Figure 6: Analysis of the average number of tool calls and average response length per round in Agentic RL.

### 4.1 WHEN FEWER TOOL CALLS LEAD TO BETTER TOOL USE

**Setup.** Based on the main experiment in section 3, we first visualize the average number of tool calls and average response length per interaction round in the training process. To further investigate the rationality and efficiency of tool usage, we filter out the correctly executed tool calling queries and calculate the average success rate of the tool calls.

**Result.** As shown in fig. 6, we identify two distinct modes in agentic reasoning: **Reactive Mode** (*short-think + frequent tool calls*) and **Deliberative Mode** (*deliberate-think + fewer tool calls*). Relating these modes to overall performance (average@32 in fig. 4), we find that the strongest models consistently adopt the Deliberative Mode, while weaker models predominantly fall into the Reactive Mode. Tool-call efficiency further explains this performance gap. As shown in fig. 8, Deliberative Mode agents achieve over 70% success in tool usage, indicating that careful reasoning before acting enables highly accurate and effective calls. In contrast, Reactive Mode agents exhibit substantially lower success rates, as their rapid, frequent calls often yield ineffective or erroneous results. Together, these findings highlight a clear *quality-over-quantity* principle: agents that invest

more inference tokens in deliberate reasoning ultimately make fewer but more successful tool calls, leading to higher efficiency of tool use and superior task performance. Thoughtful, selective tool usage thus consistently outperforms frequent but poorly targeted interactions.

---

**Takeaway 4.1: Effective mode for scaling Agentic Reasoning.**

Effective agentic reasoning follows a *quality-over-quantity* principle: investing more in deliberate internal reasoning before tool calls yields fewer but far more successful interactions, leading to higher overall efficiency and stronger performance.

---

### 4.2 INTEGRATING LONG-CoT WITH AGENTIC REASONING

**Motivation** Since investing more in deliberate internal reasoning is a more effective approach for agentic reasoning, we conduct experiments in appendix D.2 to investigate whether the long-cot models could be directly used for agentic RL. However, the results indicate that current long-cot LLMs overly rely on internal reasoning, and avoid to call the tools for reasoning tasks.

**Setup.** To address the limitation that Long-CoT models often avoid tool calls in reasoning-intensive tasks, we explicitly align them with agentic reasoning through SFT. Specifically, we leverage our SFT dataset (as described in section 2) to initialize Long-CoT models, thereby guiding them to balance deliberate internal reasoning with appropriate tool usage. This initialization enables the models to enter reinforcement learning (RL) training with a prior for effective tool invocations.

**Result.** As shown in fig. 10, the SFT-initialized Long-CoT model actively utilizes tools while retaining strong internal reasoning, demonstrating significantly improved agentic RL performance compared to the non-initialized version. However, despite this initial advantage, Long-CoT models ultimately achieve only comparable performance to instruction-based models rather than surpassing them. Analysis of response length evolution in fig. 10 reveals contrasting optimization dynamics.

**Analysis.** Instruction-based models concentrate on developing agentic reasoning capabilities from scratch without specialized internal reasoning biases, enabling continuous growth through focused tool-use learning. However, Long-CoT models face conflicting objectives: their ingrained internal reasoning patterns contradict agentic reasoning paradigms, forcing a **scaling and pruning** process where gains in agentic reasoning are offset by the need to suppress over-thinking behaviors. This dual optimization burden fragments learning efficiency, allowing instruction-based models to achieve superior scaling through concentrated capability improvement rather than divided attention between acquiring new skills and unlearning incompatible reasoning paradigms. It reveals that direct agentic RL training, where models develop reasoning and tool-use capabilities jointly from scratch, outperforms training based on Long-CoT models with conflicting internal reasoning paradigms.

---

**Takeaway 4.2: Aligning Long-CoT with Agentic RL.**

1. SFT initialization with multi-turn tool-use trajectories is essential for Long-CoT models to acquire effective tool-invocation priors before RL.
2. Instruction-based models are more suitable for agentic RL that scales the agentic reasoning ability from scratch compared to Long-CoT models with internal reasoning priors.

---

## 5 CONCLUSION

In this work, we conducted a comprehensive empirical study of reinforcement learning for agentic reasoning across the axes of **data, algorithm, and reasoning mode**. For the perspective of data curation, our findings highlight that real end-to-end multi-turn trajectories are indispensable for building strong agentic SFT foundations, while diverse and model-aware RL datasets sustain exploration and yield stable training. Algorithmically, we show that simple but effective design choices, such as clip higher, reward shaping, and token-level loss, which substantially improve training effectiveness, and that maintaining appropriate entropy is the key driver of effective agentic RL. On the reasoning side, we find a quality-over-quantity principle: fewer but more deliberate tool calls lead to superior efficiency, while Long-CoT priors often hinder tool adoption and slow down scaling. With our recipes, we effectively improve the agentic reasoning of LLMs, and we conduct a comprehensive evaluation across challenging benchmarks, including AIME2024/2025, GPQA-Diamond, and LiveCodeBench-v6, which further validates our insights.

ETHICS STATEMENT

We have conducted this research in accordance with the ICLR Code of Ethics. Our research contributes to the understanding of reinforcement learning of agentic reasoning. We recognize that the methods and findings could potentially be misused since they improve the ability of LLMs to interact and utilize external tools. Malicious use of the models and our findings may introduce uncontrollable behaviors of agents. We encourage responsible use of our contributions and recommend that practitioners consider the broader societal implications when applying these techniques.

REPRODUCIBILITY STATEMENT

To ensure reproducibility of our results, we provide comprehensive experimental details throughout this work. The appendix A.2 and appendix A.3 contain detailed descriptions of our experimental methodology, including model configurations, hyperparameters, and evaluation metrics. The curation details of the datasets are provided in appendix B.1 and appendix B.2.

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

TABLE OF CONTENTS

# A    PROBLEM FORMULATION

In this section, we first formalize the notation used throughout the paper, followed by an overview of the reinforcement learning (RL) algorithms we adopt. Finally, we outline our training and evaluation setup.

> **Research Purpose**
>
> Our goal is to *demystify reinforcement learning in agentic reasoning*. By systematically analysing three dimensions: **data**, **algorithm**, and **reasoning mode**. We extract insightful and practically applicable strategies that improve the stability, efficiency, and overall performance of agentic reasoning.

In this section, we formulate the agentic RL training objective as:

$$\max_{\pi_\theta} \ \mathbb{E}_{x\sim\mathcal{D},\, y\sim\pi_\theta(\cdot|x;\mathcal{T})}\big[\, r_\phi(x,y)\,\big] \ - \ \beta\, \mathbb{D}_{\mathrm{KL}}\big(\pi_\theta(y\mid x;\mathcal{T})\,\big\|\,\pi_{\mathrm{ref}}(y\mid x;\mathcal{T})\big). \tag{1}$$

where $\mathcal{T}$ denotes the set of available tools, $\pi_\theta$ represents the policy LLM, $\pi_{\mathrm{ref}}$ is the reference LLM, $r_\phi$ and $\mathbb{D}_{\mathrm{KL}}$ denote the reward function and KL divergence respectively. The input $x$ is sampled from dataset $\mathcal{D}$, and $y$ is the corresponding output, possibly interleaved with tool-call feedback.

Unlike conventional RL that relies purely on LLM rollouts, agentic RL (Dong et al., 2025b; Feng et al., 2025; Li et al., 2025d; Dong et al., 2025a; Singh et al., 2025) integrates tool-call feedback during reasoning. The rollout distribution factorizes as

$$P_\theta(\mathcal{R}, y\mid x;\mathcal{T}) = \underbrace{\prod_{t=1}^{t_\mathcal{R}} P_\theta(\mathcal{R}_t\mid\mathcal{R}_{<t}, x;\mathcal{T})}_{\text{Agentic Reasoning}} \cdot \underbrace{\prod_{t=1}^{t_y} P_\theta(y_t\mid y_{<t}, \mathcal{R}, x;\mathcal{T})}_{\text{Answer Generation}}. \tag{2}$$

where $\mathcal{R}$ is the reasoning trajectory of length $t_\mathcal{R}$, interleaved with tool-call feedback, and $y$ is the final answer with length $t_y$. In this paper, we mainly focus on rule-based RL algorithms like GRPO (Shao et al., 2024), which is widely adopted to optimize LLM-based Agents.

## A.1    GRPO-BASED ALGORITHM AND TECHNIQUES

Here we utilize GRPO(Shao et al., 2024) as our baseline algorithm. To better compare the difference between RL techniques that improve GRPO algorithm, we formulate the following objective in a more general format:

$$\mathcal{J}_{\mathrm{GRPO}}(\theta) = \mathbb{E}\Big[ x\sim\mathcal{D},\ \{\mathcal{R}\}_{i=1}^G \sim\pi_{\mathrm{ref}}\Big]$$

$$\mathrm{Agg}(G,\mathcal{R})\Big\{ \min\Big[ r_{i,t}(\theta)\cdot\hat{A}_{i,t},\, \mathrm{clip}\big(r_{i,t}(\theta), 1-\epsilon_{\mathrm{low}}, 1+\epsilon_{\mathrm{high}}\big)\cdot\hat{A}_{i,t}\Big] - \beta\,\mathbb{D}_{\mathrm{KL}}(\pi_\theta\,\|\,\pi_{\mathrm{ref}})\Big\}. \tag{2}$$

Here $\mathrm{Agg}(G,\mathcal{R})$ is the loss aggregation granularity, $r_{i,t}(\theta)$ is the importance ratio related to the type of loss aggregation granularity, and $\epsilon$ represents the clip ratio. $\hat{A}_{i,t}$ is the normalized advantage across all tokens:

$$\hat{A}_{i,t} = \frac{r_\phi(x, y_t) - \mathrm{mean}(\{r_\phi(\mathcal{R}_1),\dots,r_\phi(\mathcal{R}_G)\})}{\mathrm{std}(\{r_\phi(\mathcal{R}_1),\dots,r_\phi(\mathcal{R}_G)\})}. \tag{3}$$

In our empirical study, we focus on three key improvement techniques (Yu et al., 2025; Zheng et al., 2025) for GRPO: 1) **Loss Aggregation Granularity**, 2) **Reward Shaping**, 3) **Clipping Strategy**. For loss aggregation granularity, we compare two kinds of loss, which can be formulated as:

$$\mathrm{Agg}_{\mathrm{Tok}}(G,\mathcal{R}) = \frac{1}{\sum_{i=1}^G |\mathcal{R}_i|} \sum_{i=1}^G \sum_{t=1}^{|\mathcal{R}_i|}, \tag{4}$$

$$\mathrm{Agg}_{\mathrm{Seq}}(G) = \frac{1}{G}\sum_{i=1}^G, \tag{5}$$

where $\text{Agg}_{\text{Tok}}(G, \mathcal{R})$ is the token-level loss, and $\text{Agg}_{\text{Seq}}(G)$ is the sequence-level loss, and the corresponding importance ratio is formulated as:

$$r_{i,t}^{\text{Tok}}(\theta) = \frac{\pi_\theta(\mathcal{R}_{i,(t)}|\mathcal{R}_{i,<t}, \tau)}{\pi_{ref}(\mathcal{R}_{i,(t)}|\mathcal{R}_{i,<t}, \tau)} \tag{6}$$

$$r_{i,t}^{\text{Seq}}(\theta) = \left(\frac{\pi_\theta(\mathcal{R}_i|x, \tau)}{\pi_{ref}(\mathcal{R}_i|x, \tau)}\right)^{\frac{1}{|\mathcal{R}_i|}}, \tag{7}$$

here $r_{i,t}^{\text{Tok}}(\theta))$ is the importance ratio of token-level loss and $r_{i,t}^{\text{Tok}}(\theta)$ is the importance ratio of sequence-level loss. For the reward function, We optimize a composite reward that sums an outcome term (solution accuracy) and a tool-use term (number of invocations), with the tool bonus clipped to avoid degenerate "tool abuse" reward hacking. It could be formulated as:

$$r_{\text{out+tool}}(x, y, n) = \begin{cases} 1 + 0.1n & \text{if match}(y_t, y) \\ \min(-1 + 0.1n) & \text{otherwise} \end{cases} \tag{8}$$

Here, $n$ is the number of tool invocations. For another reward function, which is known as overlong reward shaping. Specifically, overlong reward shaping gives zero reward when the output length is within a safe budget, then applies a linear penalty as length approaches the maximum (from $L_{\max} - L_{\text{cache}}$ to $L_{\max}$), and assigns -1 if it exceeds $L_{\max}$. This preserves a smooth learning signal near the boundary while strongly discouraging overlong completions. It can be formulated as follows:

$$r_{\text{length}}(y) = \begin{cases} 0, & |y| \leq L_{\max} - L_{\text{cache}}, \\ \dfrac{(L_{\max} - L_{\text{cache}}) - |y|}{L_{\text{cache}}}, & L_{\max} - L_{\text{cache}} < |y| \leq L_{\max}, \\ -1, & L_{\max} < |y|. \end{cases} \tag{9}$$

## A.2 TRAINING SETUP

We choose Qwen2.5-7B-Instruct (Team, 2024) and Qwen3-4B-Instruct-2507 (Yang et al., 2025a) as our base models. For datasets, we specifically curate 3k actual agentic trajectories for SFT and 30K high-quality RL data, including math, science, and code (For more detail, please refer to section 2). For training, we employ VeRL framework, and we conduct all our experiments on $8 \times$ Tesla-A100-80G GPUs. Regarding hyperparameters for SFT, we train the base models for 5 epochs with batch size of 32, we utilize AdamW Optimizer with an initial learning rate of 5e-5, and the max response length is set to 32768. For RL training, we train 3 epochs with the batch size of 64 and the learning rate of 1e-6; the max prompt length is set to 2560. For GRPO baseline, we set the KL loss coefficient $\beta$ to 0.001, and the clip ratio $\epsilon = 0.2$ along with token-level loss aggregation, the max response length is set to 16384.

## A.3 EVALUATION SETUP

We focus on four challenging benchmarks, including AIME2024, AIME2025, GPQA-Diamond (Rein et al., 2024), and LiveCodeBench (Jain et al.). By default, we set the temperature to 1.0 and top_p to 0.6 and the maximum response length to 16384. For each problem, we sample 32 times to comprehensively evaluate average@32, pass@32, and maj@32 for AIME2024/2025 and GPQA-Diamond, for LivecodeBench, and we evaluate pass@1 and pass@5 according to its official evaluation guideline.

## B  DATASET AND EXPERIMENT SETUP

### B.1  SFT DATASET

For our self-curated real end-to-end trajectories, we utilize Qwen3-Coder-30B-A3B as the teacher model and roll out multi-turn interactions via the open-source Qwen-Agent framework with Sand-BoxFusion as the code interpreter. The SFT problems are drawn from three sources: s1-1k (Muennighoff et al.), our self-curated 3k LeetCode dataset, and a 2k ReTool multi-turn SFT set, yielding **6k**

problems in total. We generate trajectories for all 6k tasks and score the 3k LeetCode and 2k ReTool subsets using ReasonFlux-PRM (Zou et al., 2025) to filter for high-quality data. We then retain the top 1k LeetCode and 1k ReTool trajectories and keep s1-1k, resulting in a 3k real-trajectory dataset.

## B.2 RL DATASET

To investigate how diversity influences the training dynamics, we construct a diverse 30k-sample RL dataset by combining 17k DAPO-Math samples (Yu et al., 2025), 4902 math and 3586 code samples from Skywork-or1 (He et al., 2025), and 3k science problems from MegaScience (Fan et al., 2025).

## C RELATED WORK

**Tool-integrated Reasoning.** Tool-integrated reasoning (TIR) enables large language models (LLMs) to leverage external tools such as code interpreters and search engines in order to overcome the limitations of pure internal reasoning. This approach extends the computational and knowledge capacity of LLMs and allows them to tackle tasks that are infeasible through text-only reasoning. Previous TIR methods are based on prompting engineering and supervised finetuning (SFT). For example, PoT (Chen et al.), ToRA (Gou et al.), Tool-former (Schick et al., 2023), and Qwen-Math-TIR (Yang et al., 2024) train models on datasets containing tool invocation demonstrations, teaching LLMs to follow predefined patterns of tool use. Similarly, works such as ReAct (Yao et al.), MathCoder (Wang et al., 2024), and Mario (Liao et al., 2024) incorporate tool usage examples into training data so that LLMs can interleave reasoning steps with external tool calls. However, these SFT-based approaches have inherent limitations. Since models are compelled to use tools according to the distribution of training data, they cannot develop adaptive strategies for tool use, such as deciding *when* to invoke a tool, *how often* to call it, or *how to balance tool use with internal reasoning*. As a result, previous SFT-driven TIR methods improve tool-following ability but lack the flexibility and autonomy required for robust agentic behavior.

**Agent Reinforcement Learning.** Compared with SFT-based tool-integrated reasoning that merely imitates demonstrations of tool usage, it is more promising to leverage reinforcement learning algorithms (Shao et al., 2024; Guo et al., 2025; Yu et al., 2025; Liu et al., 2025a; Zhao et al., 2025) to train more capable agents. Agent Reinforcement Learning (Agent RL) explicitly models tool invocation as part of the action space and optimizes adaptive strategies through outcome-driven rewards, enabling agents to move beyond static supervision toward more flexible and effective reasoning behaviors. Representative works include search-oriented approaches such as Search-R1 (Jin et al., 2025) and R1-Searcher (Song et al., 2025), which train LLMs to interleave reasoning with search engine queries. ToRL (Li et al., 2025d) demonstrates that RL can directly optimize tool use at scale, enabling models to discover effective invocation strategies. ReTool (Feng et al., 2025) further highlights the benefit of RL by teaching models not only to use tools but also to decide when and how to call them. More recently, ZeroTIR (Mai et al., 2025) takes a complementary perspective by analyzing scaling laws in RL-based tool use, showing how strategic code invocation gradually emerges as training progresses. More general frameworks such as ARTIST (Singh et al., 2025), Tool-Star (Dong et al., 2025a), and Auto-TIR (Wei et al., 2025), further extend Agent RL to the setting of multi-tool integration. ARTIST focuses on multi-turn reasoning where agents autonomously decide not only whether to call a tool but also which tool to invoke in complex reasoning chains. Together, these frameworks highlight the frontier of Agent RL: building agents that generalize beyond single-tool domains to flexibly coordinate multiple tools under reinforcement learning objectives. However, current Agent RL methods are often tied to specific workflows and lack a systematic understanding of how reinforcement learning can more generally improve tool-use ability. Key issues such as algorithm design, tool-call efficiency, and reliance on synthetic trajectories remain underexplored.

**Entropy Mechanism for Reinforcement Learning.** Recent advances in reinforcement learning have markedly improved LLM reasoning, yet **entropy collapse**—the failure to maintain exploration ability under outcome-driven optimization, which remains a central obstacle for effective scaling of RL. At the mechanism level, Cui et al. (2025) formalizes how entropy governs exploration and identifies collapse as a key bottleneck; deepening this view, Wang et al. (2025b) show that a minority of high-entropy tokens, rather than the low-entropy majority, disproportionately drives effective learning. At the system level, He et al. (2025) provide empirical evidence that preserving entropy is essential for stable, long-horizon reasoning. Building on these insights, Cheng et al. (2025a)

and Dong et al. (2025b) incorporate entropy-aware objectives into RL to better balance exploration and exploitation in multi-turn reasoning. More recently, Deng et al. (2025) propose exploration mechanisms in Reinforcement learning with verifiable rewards (RLVR), consolidating entropy as a controllable signal, not merely a regularizer for promoting exploration, stability, and sustained improvement in agentic RL.

# D MORE EXPERIMENTS AND ANALYSIS

## D.1 MORE EXPERIMENTAL RESULTS ABOUT CLIPPING STRATEGY AND TOOL-USE EFFICIENCY

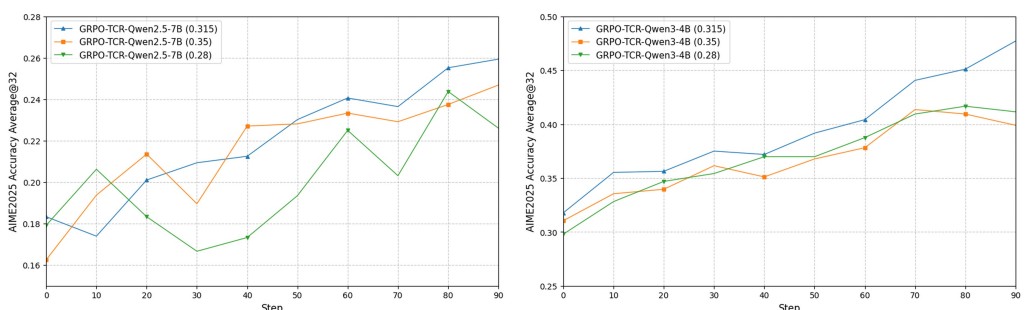

Figure 7: The analysis of clipping strategy on AIME2025 benchmark. **Left** is the analysis for Qwen2.5-7B models. **Right** is the analysis for Qwen3-4B models.

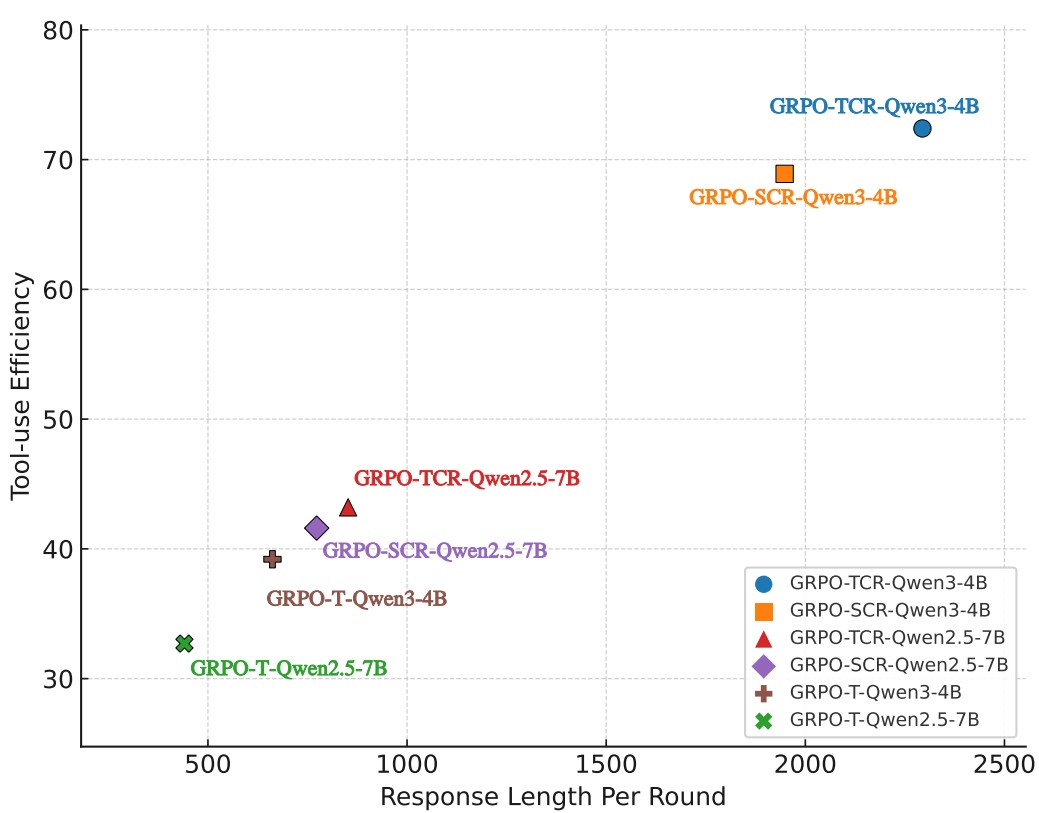

Figure 8: Tool-use efficiency comparison across different models.

## D.2 Limitations of Current Long-CoT Models in Agentic RL

**Motivation.** Motivated by our findings in section 4.1 that scaling internal reasoning before tool calls improves agentic reasoning, we explore whether incorporating Long-CoT models could further enhance agentic reasoning performance. Previous works combining Long-CoT with search engines (Search-R1 (Jin et al., 2025), R1-Searcher (Song et al., 2025) and Search-o1 (Li et al., 2025b)) have demonstrated success on knowledge-intensive tasks. Building on this foundation, we investigate whether such Long-CoT reasoning capabilities can be effectively used to benefit agentic reinforcement learning with code interpreters on reasoning-intensive problems.

**Setup.** Here we directly utilize Long-CoT LLMs like Qwen3-4B-Thinking-2507 as the starting point for RL, and utilize GRPO-TCR algorithm with the same training settings in section 3. We report the average@k and the average number of tool calls throughout training.

**Result.** As shown in fig. 9, we observed that the model achieved strong average@32 performance in the beginning, but as shown in, it hardly call the tools. As training progressed, the average number of tool calls gradually converged to zero, indicating that **Long-CoT models tend to avoid invoking tools and rely solely on internal reasoning when encountering reasoning-intensive tasks**. This behavior varies significantly by task type. For reasoning-intensive tasks, the Long-CoT models tend to utilize their internal reasoning capability to solve these tasks, thus focusing exclusively on the problem rather than analyzing the user instruction or considering calling available tools. Conversely, when confronting knowledge-intensive tasks that exceed their internal reasoning capabilities, these models could actively utilize available tools such as search engines to complete the tasks.

> **Takeaway: Limitations of Current Long-CoT models in Agentic RL**
>
> Current open-source Long-CoT LLMs optimized for reasoning tasks cannot be directly applied in Agentic RL, since they over-rely on internal reasoning and avoid invoking tools when encountering reasoning tasks.

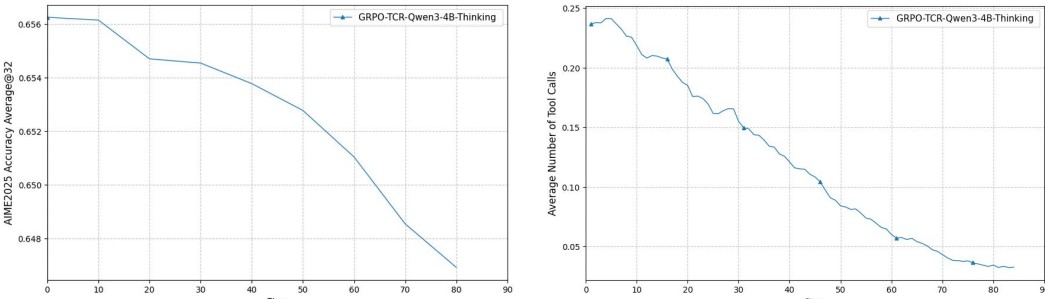

Figure 9: The training dynamics of current Long-CoT with Agentic RL.

## D.3 Comparison between Instruction-based models and Long-CoT Models

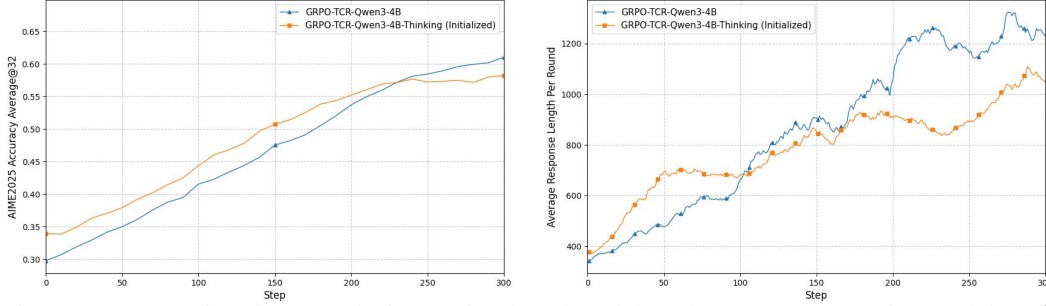

Figure 10: Comparison between the instruction-based models and Long-CoT reasoning models.**Left** is the average@32 performance on AIME2025. **Right** is the average response length during training.

## E    CONTRIBUTIONS AND COMPARISON ON CHALLENGING BENCHMARKS

Beyond the insights summarized in section 5, we contribute (i) a 3k **high-quality end-to-end agentic SFT dataset**, (ii) a 30k **diverse and effective RL dataset**, and (iii) two strong initialized cold-start model **Qwen2.5-7B-RA-SFT and Qwen3-4B-RA-SFT**, which enable broad downstream RL research, finally, (iv) empirically validated training insights that, the LLMs combined with our datasets and recipes, achieve superior agentic reasoning performance.

As demonstrated in table 2, Qwen2.5-7B-Instruct trained with our recipe significantly outperforms similar-sized models in agentic reasoning tasks. More remarkably, Qwen3-4B-Instruct-2507 trained with our approach achieves state-of-the-art performance, delivering comparable or even superior agentic reasoning capabilities compared to much larger 14B/32B models.

We distinguish between two reasoning paradigms: **Self-Contained Reasoning**, where models rely solely on their internal reasoning capabilities, and **Agentic Reasoning**, where models leverage external tools such as search engines and interpreters. Specifically, for Qwen2.5-7B-CodeAgent, we utilize model-aware data selection and train with GRPO-TCR on Qwen2.5-7B-RA-SFT, with the higher clip upper bound $\epsilon_{\text{high}} = 0.315$. For Qwen3-4B-CodeAgent, we directly utilize our full RL dataset and train with GRPO-TCR on Qwen2.5-7B-RA-SFT with the higher clip upper bound $\epsilon_{\text{high}} = 0.315$. Compared to other methods that utilize complex techniques to improve the performance of agentic reasoning, our findings proof that simple yet effective techniques could also deliver consistent improvements in agentic reasoning.

Table 2: Overall results on challenging reasoning benchmarks grouped by domain. Higher is better (%). The top two results are highlighted in **bold** and underlined.

| | MATH | | Science | Code |
|---|---|---|---|---|
| **Method** | **AIME2024** | **AIME2025** | **GPQA-Diamond** | **LiveCodeBench-v6** |
| *Self-Contained Reasoning* | | | | |
| Qwen2.5-7B-Instruct | 10.0 | 10.0 | - | 15.2 |
| Qwen3-4B-Instruct-2507 | - | 47.4 | 62.0 | **35.1** |
| Qwen2.5-72B-Instruct | 18.9 | 15.0 | 49.0 | - |
| DeepSeek-V3 | 39.2 | 28.8 | 59.1 | 3.1 |
| DeepSeek-R1-Distill-32B | 70.0 | 46.7 | 59.6 | - |
| DeepSeek-R1-Zero (671B) | 71.0 | 53.5 | 59.6 | - |
| *Agentic Reasoning* | | | | |
| Qwen2.5-7B-Instruct | 4.8 | 5.6 | 25.5 | 12.2 |
| Qwen3-4B-Instruct-2507 | 17.9 | 16.3 | 44.3 | 23.0 |
| ToRL-7B | 43.3 | 30.0 | - | - |
| ReTool-32B | 72.5 | 54.3 | - | - |
| Tool-Star-3B | 20 | 16.7 | - | - |
| ARPO-7B | 30.0 | 30.0 | - | - |
| rStar2-Agent-14B | **80.6** | 69.8 | 60.9 | - |
| *Ours* | | | | |
| **Ours (Qwen2.5-7B)** | 46.2 | 31.1 | 36.4 | 16.3 |
| **Ours (Qwen3-4B)** | 72.6 | **70.0** | **65.2** | 26 .8 |

## F    LIMITATIONS

In this work, we conduct an empirical study to investigate reinforcement learning for agentic reasoning from three key perspectives: data, algorithm, and reasoning mode. However, our experiments are conducted on small-sized models (e.g, 4B/7B). While this has already provided valuable insights into challenges and design choices for Agentic RL, recent works (Vattikonda et al., 2025) has underscores RL's extreme hyperparameter sensitivity, especially for larger-sized models. In particular, larger models may demonstrate different sensitivities to reward signals, require different exploration strategies, or exhibit more robust reasoning patterns that interact differently with RL training dynamics. We leave a more comprehensive study of RL with larger-sized models in broader agentic settings as an important future work direction.

## G    Discussion and Future Work

In this section, we outline the challenges and potential future directions of reinforcement learning in Agentic RL.

### G.1    Data-Fuel Scarcity

Based on our analysis in section 2, we believe that the training data plays a crucial role in Agentic RL, which determines both the training effectiveness and the scaling upper bound for agentic reasoning. However, for the SFT dataset that requires full end-to-end generated trajectories, it is still computationally costly to collect. Works like s1 (Muennighoff et al.) and limo (Ye et al., 2025) have demonstrated that the effectiveness of curating a small-sized but high-quality distilled dataset could significantly enhance the internal reasoning ability of LLMs. These findings suggest that we could also develop a recipe for how to curate small-sized high-quality SFT datasets, which could not only alleviate the scarcity of data in Agentic RL, but it could also improve our understanding of agentic behaviors through the insights of curating these datasets.

### G.2    Effective Scaling of Agentic Reasoning

As demonstrated in section 4, deliberate reasoning before tool invocation emerges as a superior mode for agentic problem-solving, yet effectively scaling such reasoning behaviors remains challenging. Our analysis in appendix D.2 reveals fundamental limitations in current open-source LLMs for agentic reasoning tasks. Based on that, exploring agent-specific reasoning frameworks that prioritize high-level strategic planning and efficient tool orchestration, rather than relying heavily on the model's internal reasoning capabilities, could be a promising direction for future research. Such agent-oriented reasoning chains should emphasize problem decomposition into tool-executable subtasks, strategic tool selection, and synthesis of tool outputs. This shift from reasoning-centric to high-level tool-planning-centric approaches necessitates new training methodologies and evaluation frameworks specifically tailored for agentic workflows, potentially leading to more capable autonomous agents that can navigate complex multi-step scenarios with greater reliability. We look forward to future research on exploring the compatible inference scaling methods for agentic reasoning.

### G.3    Additional Application Scenarios in Agentic Reasoning

In this work, we mainly focus on the code interpreter as a tool for agentic reasoning and find valuable insights and recipes. But we can also generalize our insights from a static and single tool environment to multi-tool and optimizable environment. For example, in this multi-tool environment, the insights of encouraging exploration in agentic RL may still hold. In a more complex environment, the correct solution to a problem consists of different possible combinations of tools, which requires more exploration for the optimal strategy and the ability to select the most effective tools for corresponding tasks.

## H    Use of Large Language Models

In this work, we utilized LLMs primarily for proofreading and language polishing to enhance readability and correct grammatical errors.

