# OpenReview forum: "Demystifying Reinforcement Learning in Agentic Reasoning"
_ICLR.cc/2026/Conference — Submitted to ICLR 2026_

### Official Review · Reviewer_f21X · 2025-10-16

**Soundness:** 2
**Presentation:** 3
**Contribution:** 2
**Rating:** 4
**Confidence:** 3

**Summary:**

This paper presents a systematic empirical study on reinforcement learning for agentic reasoning in large language models. It investigates three central aspects: data, algorithm, and reasoning mode. The authors find that real end-to-end multi-turn tool-use trajectories provide a stronger foundation than synthetic data, while diverse and model-aware reinforcement learning datasets help maintain exploration and training stability. Algorithmically, the study shows that using higher clipping thresholds, overlong reward shaping, and token-level loss aggregation improves training efficiency and final performance. Maintaining balanced entropy is identified as essential for stable optimization. In reasoning strategy, models that engage in deliberate reasoning with fewer but more accurate tool calls achieve higher efficiency and accuracy compared to those relying on frequent tool use. The work offers clear empirical insights and practical guidelines for improving agentic reasoning, achieving strong results on challenging benchmarks including AIME2024, AIME2025, GPQA-Diamond, and LiveCodeBench-v6.

**Strengths:**

1. The goal to demystify agentic RL by isolating data algorithm and reasoning mode decisions is explicit and well structured

2. The paper is organized and domenstrated properly and clearly.

3. This paper provides useful empirical insights
- Real end to end trajectories substantially outperform stitched synthetic trajectories at SFT with large gains on AIME metrics
- Dataset diversity and model awareness are shown to affect entropy exploration and learning dynamics not just end metrics
- The analysis of deliberative versus reactive tool use connects behavioral telemetry tool call frequency and success rates to accuracy

**Weaknesses:**

1. Although the article provides a series of insights into the application of RL in LLM agentic reasoning, the corresponding conclusions are often obtained only through simple task performance comparisons and qualitative analysis, lacking in-depth analysis of the causes or underlying mechanisms.

2. In Section 2.1, the authors claim that real end-to-end trajectories are superior to synthetic stitched ones. However, the performances of the two trajectory types appear relatively similar, and generating a large number of real trajectories requires a teacher model, suggesting that the model's capabilities may be limited by those of the teacher model.

2. While the paper compares real versus synthetic trajectories and diverse versus math only datasets it is not always clear which specific factors within these bundles drive the gains for example trajectory length tool type distribution or supervision quality. The overlong shaping and clip higher are bundled with other changes when moving from GRPO-T to GRPO-TCR which can blur attribution

**Questions:**

1. Will the real trajectory generated by the teacher model limit its ability ceiling?

2. Could the authors provide detailed information about the two datasets used in Section 2.2 to demonstrate that the performance differences are not caused by other factors, such as quality, size, etc.?

3. The paper draws several key conclusions: on one hand, external tools can introduce novel knowledge to enhance the effectiveness of reinforcement learning; on the other hand, frequent tool invocations without adequate reasoning may result in inefficiencies, while prolonged internal thinking could suppress the model's propensity to invoke tools. What might potential breakthroughs in this area entail?

---

### Official Review · Reviewer_Lr62 · 2025-10-29

**Soundness:** 2
**Presentation:** 2
**Contribution:** 2
**Rating:** 2
**Confidence:** 4

**Summary:**

This paper presents a systematic empirical investigation into optimizing LLMs for agentic reasoning using RL. The authors explore three key dimensions: data, algorithm, and reasoning mode. Through experiments on models like Qwen2.5-7B and Qwen3-4B, the study yields several key insights and takeways, such as the diversity of dataset and exploration-oriented RL techniques. The paper demonstrates that applying these recipes significantly enhances agentic reasoning, allowing smaller models to achieve performance comparable or superior to larger models on challenging benchmarks. The authors also contribute curated datasets.

**Strengths:**

1. This paper investigates a crucial research problem of finding the optimal recipes for agentic reasoning for LLMs.
2. New datasets and training strategies are proposed which are shown to be effective on typical benchmarks.
3. The paper distills its experimental results into concise and practical takeaways, which provide clear recommendations for practitioners working on agentic RL.

**Weaknesses:**

1. The interplay between different factors (data, algorithm, reasoning mode) is not clear and requires further experiments and analysis.
2. The protocol for selecting experimental LLMs is not stated.
3. As the paper is a comparative study, using only two LLMs for experiments are insufficient to prove the universality of the findings.
4. Related works is provided in separated sections. Considering the popularity of this field, a comprehensive discussion on the relationship between this work and related works is highly needed.
5. The paper identifies effective recipes, but the ingredients are largely drawn from prior work, which questions the algorithmic novelty of this work.

**Questions:**

Please see weaknesses.

---

### Official Review · Reviewer_yVMD · 2025-10-30

**Soundness:** 3
**Presentation:** 1
**Contribution:** 3
**Rating:** 4
**Confidence:** 4

**Summary:**

This paper empirically investigates key design choices for applying Reinforcement Learning (RL) to enhance Large Language Model (LLM) agentic reasoning across three areas: data, algorithms, and reasoning mode. The study discovers that using real end-to-end tool-use trajectories and diverse, model-aware datasets is crucial for robust SFT initialization and sustained RL exploration. Furthermore, the authors find that exploration-friendly RL algorithms and a deliberative, low-frequency tool-call strategy consistently lead to improved efficiency and final accuracy. The paper's overall contribution is a collection of recipes that enable models to achieve superior results on challenging agentic benchmarks.

**Strengths:**

- The studies in this paper would be useful for practitioners seeking prescriptive guidance on how to get RL-based agentic reasoning systems running in practice. It provides a collection of empirical recipes and observations that, while not cohesively unified, offer concrete starting points for implementation.

- The “Algorithm-wise” motivation in the introduction is well articulated and closely tied to the paper’s stated goal of demystifying how different RL design choices affect agentic reasoning. It identifies a real gap in current understanding and sets up the most coherent through-line in the paper.

- The paper highlights several practically valuable directions for advancing RL in agentic reasoning, including the use of real, end-to-end trajectories for stronger initialization, model-aware datasets that match task difficulty to model capacity, and careful analysis of how clipping and entropy settings affect training stability. These insights are well motivated and provide concrete guidance for practitioners seeking to improve the robustness and efficiency of RL-based reasoning systems.

**Weaknesses:**

While I enumerate some specific issues below, at a high level this paper reads as a loose compilation of observations rather than the cohesive study suggested by its title and abstract (“demystifying RL in agentic reasoning”). The work follows a repeated motivation-setup-result–analysis pattern across ~6 pages, but the abrupt transitions between sections prevent any sustained narrative flow. Although empirical studies are valuable, the chosen structure makes the paper difficult to read, evaluate as a coherent whole, and get in depth insights from. As a result, the paper reads like an extended ablation study without a clear overarching research question or conceptual synthesis despite some efforts in the introduction to frame it. The authors clearly have an in-depth understanding of what in agentic reasoning requires demystifying, but their work here does a poor job in fulfilling their goals. This work would greatly improve if the authors were to focus on narrative structure, focus on two or three insights in depth instead of the many in this work, and avoid presenting every observation as a major takeaway. Furthermore, the frequent use of arbitrary bolding, numerous typos, and awkward phrasing distract from the main ideas. More detailed examples are provided below.

- Several claims in the introduction are unsubstantiated. The statement about a “shift” enabling RL progress lacks any supporting references, and the point about unstable training dynamics relies on only one citation for a single failure mode. Without stronger evidence or broader literature context, these arguments feel underdeveloped and weaken the paper’s motivation.

- The paper contains numerous grammatical errors and awkward phrasings that make it difficult to read. Some of the early examples are enumerated in the following sentences, but a substantial editing pass over this whole work is required. The final abstract sentence (“could also achieve”) should use “do achieve” unless this is an unsubstantiated hypothesis and in that case I would recommend removing it. The first sentence of point (3) in the introduction is very awkward/clunky phrasing. Having multiple typos such as “algoithm,” “cliping,” “Exploiatation,” and “Tarajectories” in the main figure (Figure 1), which is likely the first and most visible element readers encounter, is unacceptable and reflects a lack of proofreading and care prior to submission. Furthermore, the first organizational bullet under the introduction (“we curate real end-to-end SFT dataset and high-diversity, model-aware RL dataset that improve”) is grammatically incorrect. Similarly, “clipping and KL divergence penalty” should use the plural form “penalties.”

- The bolding throughout this paper is quite heavy and often distracting. For instance, GRPO-TCR is bolded inconsistently and seemingly without reason in Section 3.1 when compared to the rest of the paper, and some later lines (e.g., line 460) are bolded without clear purpose. When large portions of text are bolded (as is done in Section 3.1), it undermines emphasis entirely. The overall formatting should be cleaned up and standardized substantially. Likewise, the bolding of the “R” in Reward and “C” in Clip in 3.1 appears intended to link these terms narratively to variants of GRPO (e.g., TCR). However, this formatting choice seems unnecessary and distracting, especially given the heavy usage of bolding throughout the paper. It would be clearer to simply and directly state which elements of the acronyms correspond to which components rather than implying it through typography.

**Questions:**

The abstract claims that the proposed recipes “boost agentic reasoning ability across four benchmarks” and enable (could enable?) 4B models to outperform 32B models is not directly substantiated by the presented experiments. The paper demonstrates improvements in individual ablations but does not report a unified evaluation showing all insights combined outperforming prior baselines. Am I missing an experiment where multiple design factors are jointly applied to validate this claim?

---

### Official Review · Reviewer_Xvk9 · 2025-11-10

**Soundness:** 3
**Presentation:** 2
**Contribution:** 3
**Rating:** 4
**Confidence:** 4

**Summary:**

This paper presents an empirical study of reinforcement learning (RL) for agentic reasoning in LLMs, focusing on Qwen2.5-7B and Qwen3-4B models. The authors systematically analyze three factors: data curation, algorithmic design, and reasoning mode. On the data side, they compare real end-to-end tool-use trajectories with synthetic stitched trajectories, and construct a diverse, model-aware RL dataset. Algorithmically, they study GRPO-based variants with different clipping strategies, loss aggregation granularities, and reward shaping. On the reasoning side, they investigate how the number of tool calls and response length affect performance and identify a “deliberative” regime with fewer but more accurate tool calls. They evaluate on AIME2024, AIME2025, GPQA-Diamond, and LiveCodeBench-v6, and also release an agentic SFT dataset and an RL dataset.

**Strengths:**

- Comprehensive empirical analysis across three key axes: data (real vs synthetic trajectories, diversity, model-aware selection), algorithm (clipping, reward shaping, loss granularity), and reasoning mode (tool-call frequency vs internal reasoning).

- The paper considers both aligned and Long-CoT model families and multiple model sizes, which makes the conclusions more broadly informative.

- The empirical takeaways (e.g., benefits of real end-to-end SFT data, importance of maintaining entropy, and advantages of more deliberate tool usage) are concrete and potentially useful for practitioners.

- The paper includes the construction of real end-to-end agentic SFT data and an RL dataset for agentic reasoning, which are valuable resources if they are released as claimed.

**Weaknesses:**

- The algorithmic study around GRPO feels somewhat incomplete. A vanilla GRPO-S (sequence-level loss) baseline would be a natural reference point to compare against GRPO-T, GRPO-TCR, and GRPO-SCR.

- Similarly, the contribution of each component in the improved recipes is hard to isolate. For example, GRPO-TR / GRPO-SR (only overlong reward shaping) or GRPO-TC / GRPO-SC (only higher clipping) would help decouple the effect of overlong reward shaping from that of the more permissive clipping.

- The plots seem to be based on single runs. Showing the mean and standard error (or confidence intervals) over multiple seeds would make the statistical significance of the claims more convincing.

**Questions:**

1. In the construction of the model-aware dataset for Qwen2.5 based on the Qwen3 difficulty histogram:
How exactly is the filtering and rebalancing done? From the description it seems you discard “trivial” problems with 0% and 100% accuracy, but do you do anything else beyond this filtering?

2. For Long-CoT models and agentic RL:
Did you try other GRPO variants (GRPO-T, GRPO-S, GRPO-SC, GRPO-TC, GRPO-SR, GRPO-TR) specifically in the Long-CoT setting? The paper concludes that these methods are not suitable for agentic RL for Long-CoT models, but the experiments appear to focus on one particular RL pipeline/recipe. Given that the paper itself emphasizes how sensitive agentic RL is to algorithmic design, it would be helpful to see at least a brief investigation of these variants, or a clearer justification for why they are expected not to help.

---

### Meta-Review · Area_Chair_KB69 · 2025-12-12

**Summary:**

All reviewers are negative and the authors do not provide any response. I, therefore, recommend reject.

**Reviewer Scores:**

N/A

---

### Decision · Program_Chairs · 2026-01-26

Reject